# Polymorphic Covalent Organic Frameworks: Molecularly Defined Pore Structures and Iodine Adsorption Property

**DOI:** 10.3390/molecules28010449

**Published:** 2023-01-03

**Authors:** Canran Wang, Shan Jiang, Wenyue Ma, Zhaoyang Liu, Leijing Liu, Yongcun Zou, Bin Xu, Wenjing Tian

**Affiliations:** 1State Key Laboratory of Supramolecular Structure and Materials, College of Chemistry, Jilin University, Changchun 130012, China; 2College of Chemistry and Chemical Engineering, Liaoning Normal University, Dalian 116029, China; 3State Key Laboratory of Inorganic Synthesis and Preparative Chemistry, College of Chemistry, Jilin University, Changchun 130012, China

**Keywords:** covalent organic framework, polymorphism, iodine adsorption

## Abstract

Radioactive iodine-capturing materials are urgently needed for the emerging challenges in nuclear waste disposal. The various pore structures of covalent organic frameworks (COFs) render them promising candidates for efficient iodine adsorption. However, the detailed structure–property relationship of COFs in iodine adsorption remains elusive. Herein, two polymorphic COFs with significantly different crystalline structures are obtained based on the same building blocks with varied molecular ratios. The two COFs both have high crystallinity, high specific surface area, and excellent chemical and thermal stability. Compared with the [C_4_+C_4_] topology (PyT-2) with an AA stacking form, the [C_4_+C_2_] topology (PyT-1) with an AB stacking form has more twisted pore channels and complex ink-bottle pores. At ambient conditions, PyT-1 and PyT-2 both exhibit good adsorption properties for iodine capture either in a gaseous or liquid medium. Remarkably, PyT-1 presents an excellent maximum adsorption capacity (0.635 g g^−1^), and the adsorption limit of PyT-2 is 0.445 g g^−1^ in an *n*-hexane solution with an iodine concentration of 400 mg L^−1^, which is highly comparable to the state-of-the-art iodine absorption performance. This study provides a guide for the future molecular design strategy toward novel iodine adsorbents.

## 1. Introduction

With the emerging employment of nuclear energy, tremendous attention has been paid to the proper disposal and treatment of radioactive pollutants [1,2,3]. The ^131^I and ^129^I nuclides produced through uranium fission are highly volatile and hazardous. ^131^I has high radioactivity (4.6 × 10^15^ Bq·g^−1^) and high activity that can participate in human metabolism [4], and the half-life of ^129^I is as long as 1.57 × 10^7^ years, which is the major pollutant endangering human health and ecological environment [5,6]. The reprocessing condition of nuclear fuels usually requires a high temperature under high pressure. As-produced radioactive iodine exists in both vapor and liquid forms. For iodine vapor, the main capturing strategy is through physisorption [7,8,9,10,11]. Bismuth-based compounds [12,13,14] and mordenite-containing silver [11,15,16] are usually used to adsorb gaseous iodine. However, these adsorbents usually have small porosity and limited specific surface area; thus, it is difficult to achieve the efficient adsorption of gaseous iodine [17]. Recently, metal–organic frameworks (MOFs) have been employed to improve porosity and specific surface area. However, MOFs cannot be applied in liquid media, which require a wet-washing process due to their poor water stability [18,19]. By comparison, covalent organic frameworks (COFs) with porous structures and high stability have been demonstrated as potential adsorbents for iodine in vapor and various liquid media [20,21,22,23,24,25,26,27,28,29,30,31,32,33,34,35].

The mechanism of iodine adsorption for COF materials involves both physisorption and chemisorption, with physisorption playing the dominant role. For physisorption, the pore size and morphology of organic porous materials have important effects on the capture of iodine [10]. In general, most COFs have two-dimensional AA stacked topology, resulting in dense and uniformly distributed one-dimensional pore channels. This “clean and perfectly straight” pore structure of COFs normally leads to a possible desorption process, i.e., the desorption and adsorption processes occur simultaneously, which limits the further improvement of the maximum iodine adsorption capacity. To address this challenging issue, it is, therefore, necessary to adjust the pore structure and morphology of COFs by controlling the stacking form [36,37,38,39,40] or introducing flexible building blocks [41,42,43,44]. In this case, the twisted pore structure, rather than the perfectly straight structure, can efficiently expose the adsorption site while hindering the desorption process of iodine molecules, thus leading to the maximum adsorption capacity [42,43,45,46]. Therefore, it is highly appealing to design a COF with a molecularly defined pore structure for efficient iodine adsorption.

Herein, we report a precisely tuned pore structure of COFs by controlling the molecular ratio between the building blocks and the subsequent stacking orders. Monomer 4,4′,4″,4″′-(pyrene-1,3,6,8-tetrayl)tetrabenzaldehyde with a large conjugated structure and monomer N,N,N′,N′-tetrakis(4-aminophenyl)-1,4-benzenediamine with rich nitrogen elements were selected to facilely construct two-dimensional imine-linked COFs PyT-1 and PyT-2 with different topology combinations and stacking forms. Both PyT-1 and PyT-2 showed large specific surface area, high chemical, and thermal stabilities, as well as high iodine adsorption capacity. PyT-1 with an AB stacking form exhibited an excellent iodine adsorption property of 0.635 g g^−1^ in an n-hexane solution with a concentration of 400 mg L^−1^, which is comparable to the state-of-the-art iodine-absorbing materials reported so far. These results illustrate the significant variation in the iodine-capturing properties based on the molecularly defined pore structure, which may further stimulate the design strategy of novel iodine absorbents and their related applications.

## 2. Results and Discussion

Two COFs were synthesized via Schiff-base condensation reaction between 4,4′,4″,4″′-(pyrene-1,3,6,8-tetrayl)tetrabenzaldehyde (TFPPY) and N,N,N′,N′-tetrakis(4-aminophenyl)-1,4-benzenediamine (TPDA) (Figure 1). As a typical procedure, a mixture of TFPPY and TPDA in two solvent systems was heated in a sealed Pyrex tube at 120 °C for 7 days. PyT-1 was obtained by using 1,4-dioxane-mesitylene-acetic acid (aq., 6 M) (5/5/1, *v*/*v*/*v*) mixtures as the solvents, and PyT-2 was synthesized by using benzyl alcohol (MeOH)-acetic acid (aq., 6 M) (10/1, *v*/*v*) mixtures as the solvents.

As shown in Figure 1A, solid-state ^13^C NMR presented a signal peak at 154 ppm for PyT-1 and 156 ppm for PyT-2, which corresponded to the C atom on the C=N imine bond. The structures of the COFs were further verified using Fourier-transform infrared (FTIR) spectra. As shown in Figure 1B (blue curve and pink curve), the appearance of the C=N stretching vibration peak at 1622 cm^−1^ verified the occurrence of the imine condensation reaction. The infrared spectra and the NMR spectra of the two COFs were almost identical, indicating that the chemical structures of the two COFs were nearly the same.

The crystalline structure of the two COFs was analyzed via powder X-ray diffraction (PXRD), structural simulation, and Pawley refinement. The experimental profile (black curve in Figure 2A) was consistent with the simulated PXRD pattern for the staggered (AB) stacking form of PyT-1 (pink curve in Figure 2A) but was different from the eclipsed (AA) stacking form (Appendix A), demonstrating that the crystalline structure of PyT-1 had an AB stacking form (Figure 2B,C). The PXRD pattern for the [C_4_+C_2_] topology combination (pink curve in Figure 2A) displayed distinctly intensive PXRD peaks at 2.68°, 5.12°, 6.00°, 8.04°, and 10.96°, corresponding to (100), (200), (210), (220), and (400), indicating the high crystallinity of PyT-1. The lattice parameters were optimized using Pawley refinement until the error converged within a certain range (R_wp_ = 3.03%, R_p_ = 6.08%, respectively). The lattice parameters were a = 33.90Å, b = 56.32Å, c = 9.28Å, α = 89.73°, β = 89.20°, and γ = 92.36°, respectively.

The experimental profile (black curve in Figure 2D) was consistent with the simulated PXRD pattern for the AA stacking form of PyT-2 (blue curve in Figure 2A) but was different from the AB stacking form (Appendix A), demonstrating that the crystalline structure of PyT-2 had an AA stacking form (Figure 2E,F). The PXRD pattern for the [C_4_+C_4_] topology combination (blue curve in Figure 2D) displayed distinctly intensive PXRD peaks at 5.46°, 7.68°, 10.94°, and 12.28°, corresponding to (110), (200), (220), and (310), indicating the high crystallinity of PyT-2. The lattice parameters were optimized through Pawley refinement until the error converged within a certain range (R_wp_ = 3.32%, R_p_ = 4.23%, respectively). The lattice parameters were a = 7.67Å, b = 21.66Å, c = 24.53Å, α = 89.51°, β = 88.17°, and γ = 91.82°, respectively. The different monomer connection modes and stacking forms of PyT-1 and PyT-2 resulted in the different crystalline structures of the two COFs. PyT-1 with the [C_4_+C_2_] topology combination and the AB stacking form revealed a very staggered skeleton and distorted pores (Figure 2B,C), while the pore structure of PyT-2 with the [C_4_+C_4_] topology combination and the AA stacking form was uniform (Figure 2E,F).

SEM and TEM images showed that PyT-1 exhibited a randomly distributed nanosheet morphology with an average diameter of 200 nm (Figure 3A,C and Appendix A), while PyT-2 exhibited a distinct tubular morphology, which was an ordered stacking with a nanosheet structure observed from the periphery (Figure 3B and Appendix A). TEM images clearly demonstrated the hollow nature and rough surface of the tubes (Figure 3D).

To characterize the porosity of the COFs, an experiment was performed using nitrogen adsorption isotherms. The two COFs exhibited the typical II adsorption isotherms. However, a convex curve appeared in PyT-1 at a relatively lower pressure of 0.01 < P/P_0_ < 0.1 (Figure 4A), indicating its porosity and a combination of type-I nitrogen sorption isotherm and type-H_2_ hysteresis loops. The BET surface area and pore volume of the PyT-1 were remarkably high, achieving 1186.94 m^2^ g^−1^ and 0.89 cm^3^ g^−1^, and those values for PyT-2 were 704.96 m^2^ g^−1^ and 0.65 cm^3^ g^−1^ (P/P_0_ = 0.99), respectively. The pore size of PyT-2 was calculated to be mainly around 0.8 and 1.0 nm by using the non-local density functional theory (NLDFT). Additionally, the pore size of PyT-1 was mainly around 1.09 nm, while some pores were distributed around 2.02 nm (Figure 4B). The coexistence of micropores and sub-mesopores demonstrated the unique hierarchical porous structure of PyT-1, which contained ink-bottle-shaped pores and showed potential application in adsorption.

Thermogravimetric analysis (TGA) was performed to evaluate the thermal stability of COFs. PyT-1 and PyT-2 were stable before 350 °C. Even at 560 °C, the weight loss of the pristine COFs still remained below 20% (Appendix A), which showed good thermal stability. The COFs were also stable in organic solvents. We immersed the COF samples in MeOH, cyclohexane (CYH), tetrahydrofuran (THF), N,N-dimethylformamide (DMF), H_2_O, 1M HCl, and 1M NaOH solvents for 24 h, and the COFs maintained their chemical structures (Appendix A).

In order to highlight the iodine adsorption property, we first performed iodine vapor adsorption tests by exposing the COFs to a temperature of 75 °C under ambient pressure. The maximum adsorption capacity *q* toward iodine was calculated using Equation (1):*q* = (*m*_2_ − *m*_1_)/*m*_1_(1)
where *q* is the adsorption value of iodine (g g^−1^), and *m*_1_ and *m*_2_ are the masses of the COFs before and after the uptake of iodine, respectively.

As shown in Figure 5A, the amount of I_2_ adsorption significantly increased within the initial 10 h and then slowly raised afterward. After 90 h, no obvious change was observed, indicating the arrival of adsorption equilibrium. The adsorption limit of the iodine vapor reached 3.82 g g^−1^ and 3.41 g g^−1^ for PyT-1 and PyT-2, respectively. After being exposed to the air at room temperature for seven days, more than 97% of the iodine molecules were retained in the COFs (Appendix A). The iodine-laden COFs were rinsed in ethanol to release the adsorbed iodine for reuse in the next cycle. The iodine-captured PyT-1 sample was recyclable through ethanol rinse, retaining a 3.46 g g^−1^ adsorption limit after five cycles. Under the same conditions, PyT-2 retained a 2.62 g g^−1^ adsorption limit after five cycles. Moreover, the structural stability of PyT-1 and PyT-2 against organic solvents was also demonstrated by the iodine vapor adsorption experiment.

As shown in Appendix A, the adsorption capacities of the COFs exhibited neglectable variations before and after immersing them in *n*-hexane, suggesting their maintained pore structures without degradation.

In order to explore the adsorption capacity of the COFs in the organic solution system, we conducted experiments with iodine adsorption solutions. The experiments were performed in *n*-hexane at room temperature. Iodine-containing solutions at concentrations of 400 mg L^−1^ and 200 mg L^−1^ were prepared. The residual iodine concentration was measured at 523 nm with an ultraviolet spectrophotometer (Appendix A). The adsorption limit *q* of PyT-1 reached up to 0.365 g g^−1^ in 200 mg L^−1^ of iodine solution, and the adsorption limit *q* of PyT-2 reached up to 0.315 g g^−1^ (Appendix A). The adsorption limit *q* of PyT-1 reached up to 0.635 g g^−1^ in 400 mg L^−1^ of the iodine solution, and the adsorption limit *q* of PyT-2 reached up to 0.445 g g^−1^ (Figure 6). It can be seen that the adsorption value rapidly increased in the first 40 min, and the adsorption equilibriums were reached within 180 min. The pseudo-first-order and pseudo-second-order kinetic models were used to explore the adsorption kinetics. The experimental data fitted well to the pseudo-second-order adsorption kinetics model with a good correlation coefficient (R^2^ = 0.998), demonstrating that the adsorption process of PyT-1 to iodine followed the pseudo-second-order kinetic model.

## 3. Materials and Methods

### 3.1. Materials

TFPPY and TPDA were supplied by Jilin Chinese Academy of Sciences, Yanshen Technology Co. Ltd. (Changchun, China), and Shanghai Maclean Biochemical Technology Co. Ltd. (Shanghai, China); 1,4-Dioxane, acetic acid, mesitylene, MeOH, DMF, ethanol, THF, CYH, *n*-hexane, and acetone were supplied by Sigma-Aldrich and Aladdin (Shanghai, China). Iodine (99.8%) ^127^I was purchased from Anhui Zesheng Technology Co., Ltd. (Anhui, China), which is non-radiologic instead of radiologic iodine due to having the same physical and chemical properties. Other chemical reagents were commercially available. Unless otherwise indicated, all the reagents were used without further purification.

### 3.2. Synthesis of PyT-1 and PyT-2

COFs were synthesized using the solvothermal method. Typically, TFPPY (24.7 mg, 0.04 mmol), TPDA (9.45 mg, 0.02 mmol), and two different solvent systems, one as a mixture of 1,4-dioxane and mesitylene (1/1, 1 mL) and the other as MeOH (1 mL), were mixed in a Pyrex tube (10 mL) and sonicated for 10 min to obtain a homogeneous dispersion solution. After sonication for 10 min, 0.1 mL of 6 M aqueous acetic acid was added to the solution. The tube was degassed through the typical three freeze–pump–thaw cycles with liquid nitrogen, and it was sealed with butane flames and then heated at 120 °C for 7 days. The dark red precipitate was collected through filtration and rinsed with DMF, THF, and acetone. The precipitates were collected and vacuum-dried at 120 °C for 4 h.

### 3.3. Characterizations

Powder X-ray diffraction (PXRD) was performed using a Riguku DMAX2550 diffractometer (Tokyo, Japan) using Cu-Kα radiation, 40 kV, 200 mA at room temperature. Fourier-transform infrared (FTIR) spectra were recorded on a Vertex 80V spectrometer. The solid-state ^13^C cross-polarization/magic-angle spinning (CP/MAS) NMR spectra (CA, USA), were collected with a VARIAN Infinity plus 400 spectrometer. Scanning electron microscopy (SEM) images (Tokyo, Japan) were obtained using a JEM-6700F microscope (JEOL). Transmission electron microscope (TEM) images (Tokyo, Japan) were obtained using a JEM-2100F microscope (JEOL). The samples were prepared through drop-casting PyT-1 and PyT-2 aqueous dispersion over a copper grid. The nitrogen adsorption–desorption isotherms of the COFs were measured on a Micromeritics ASAP 2020 instrument (GA, USA). Before measurement, the samples were degassed under vacuum at 120 °C for more than 4 h. The pore width distribution was acquired using the non-local density functional theory (NLDFT) model. TGA was performed at a heating rate of 10 °C min^−1^ from room temperature to 800 °C in a N_2_ atmosphere by using a Q500 thermal analyzer system. The absorbance of iodine was measured at 523 nm with an Analytic Jena Specord 210 ultraviolet spectrophotometer.

## 4. Conclusions

By varying the synthesis conditions, two polymorphic two-dimensional covalent organic frameworks with different crystal structures were synthesized using the same monomers. The two obtained COFs both revealed high crystallinity, high specific surface area, high stability, and good iodine adsorption capacity in vapor and liquid media. The adsorption limit for PyT-1 with AB accumulation and distorted pores in an *n*-hexane iodine solution was remarkably high, i.e., 0.635 g g^−1^, which is comparable to the state-of-the-art iodine adsorption performance. This research provides a guide for the molecular-design strategy of efficient iodine-adsorbing materials used in environmental fields.

## Data Availability

The data presented in this study are available on request from the corresponding author.

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
