# Peer review of "Polymorphic Covalent Organic Frameworks: Molecularly Defined Pore Structures and Iodine Adsorption Property"

_molecules, 2023, doi:10.3390/molecules28010449_

Round 1

Reviewer 1 Report

Wang et al. constructed two COFs with different crystalline structures from the same building blocks via regulating molecular ratios. The two COFs were characterized in detail and both of them possessed good adsorption property for iodine capture. The manuscript could be suitable for Molecules after consideration of the following points:

1.     Figure 1A should be in front of Figure 1B.

2.     The “isomers” usually referring to molecules have the same molecular formula and chemical composition but exhibit different spatial atomic arrangement. However, the chemical compositions of the two COFs in this manuscript are not the same, even though they are synthesized from the same building units. The authors should carefully consider the accuracy of this claim.

3.     Some literature about constructing COFs with different structures via tuning molecular ratios should be cited, such as “Chem Mater, 2018, 30, 1762–1768”, “Sci China Chem, 2022, 65, 190–196”

Author Response

Response to Reviewer 1 Comments

We would like to thank Reviewer #1 for the positive assessment and recommendation for publication of our work. We have now considered all the constructive comments and made the requested revisions accordingly.

R1_Q1: Figure 1A should be in front of Figure 1B.

Answer to R1_Q1: We have now rephrased the caption of Figure 1A and 1B.

R1_Q2: The “isomers” usually referring to molecules have the same molecular formula and chemical composition but exhibit different spatial atomic arrangement. However, the chemical compositions of the two COFs in this manuscript are not the same, even though they are synthesized from the same building units. The authors should carefully consider the accuracy of this claim.

Answer to R1_Q2: We thank the Reviewer for noticing this inaccuracy in our description. We agree that since the chemical compositions of the two COFs in this manuscript are not the same, they should not be addressed as isomers. We have now revised the titleto “Polymorphic covalent organic frameworks: molecularly defined pore structures and iodine adsorption property”. The corresponding modification has also been indicated in the article.

R1_Q3: Some literature about constructing COFs with different structures via tuning molecular ratios should be cited, such as “Chem Mater, 2018, 30, 1762–1768”, “Sci China Chem, 2022, 65, 190–196”

Answer to R1_Q3: We would like to thank the Reviewer for this kind suggestion. Now the corresponding references have been cited in the revised manuscript.

Reviewer 2 Report

Here the authors have reported two well-tuned COF structures by controlling the molecular ratio of the building blocks. The new structures are PyT-1 and PyT-2, fully characterized by various methods. These new MOFs have shown good iodine uptake ability from both vapour phase and liquid phase. The work is suitable for the journal molecule with minor corrections.

1. TGA and PXRD are sometimes confusing while stating stability. A COF might be degraded way before without showing a mass loss. Similarly, in PXRD 50% retention of structure can lead to the same pattern. One should try the adsorption isotherm of these materials after immersing them in the solvents.

Reviewer 3 Report

Tian et al. reported Py-T COFs for radioactive wastes I2 adsorption. This is an interesting and important work. I suggest it to be accepted only after the following minor revision.

Some important references for covalent organic frameworks are suggested to be cited, such as “Frontiers in Chemistry, 2019, 7, 636”, “Science China Chemistry, doi:10.1007/s11426-022-1350-1”, “Materials, 2021, 14, 5600” and “Molecules, 2020, 25, 2425”.

Author Response

Response to Reviewer 3 Comments

We would like to thank Reviewer 3 for the positive assessment of our manuscript, especially for classifying our work as “interesting” and “important”.

R3_Q1: Some important references for covalent organic frameworks are suggested to be cited, such as “Frontiers in Chemistry, 2019, 7, 636”, “Science China Chemistry, doi:10.1007/s11426-022-1350-1”, “Materials, 2021, 14, 5600” and “Molecules, 2020, 25, 2425”.

Answer to R3_Q1: We would like to thank the Reviewer for this insightful suggestion. Now the corresponding references have been cited in the revised manuscript.